
# Earthquake Response of Heavily Damaged Historical Masonry Mosques after Restoration

Ahmet Can Altunışık[1], Ali Fuat Genç[1]

[1]Department of Civil Engineering, Karadeniz Technical University, Trabzon, 61080, Turkey

*Correspondence to*: Ahmet Can Altunışık (ahmetcan8284@hotmail.com)

**Abstract.** Restoration works have been accelerated substantially in Turkey in the last decade. Many historical buildings, mosques, minaret, bridges, towers etc. structures are restored. With these restorations an important issue arises, namely how restoration work affects the structure. For this reason it is aimed to investigate the restoration effect on the earthquake response of a historical masonry mosque with considering the openings on masonry dome. For this purpose, Hüsrev Pasha Mosque which is located in Ortakapı district
in old city of Van, Turkey. The region of Van is in an active seismic zone therefore earthquake analyses were made in this study. Firstly finite element model of the mosque was constructed considering the restoration drawings and 16 window openings on dome. Then model was constructed with 8 window openings. Structural analyses were made under dead and earthquake loads and Mode Superposition Method was used in analyses. Maximum displacements, maximum-minimum principal stresses and shear stresses are given with contours diagrams. The results are investigated according to Turkish Earthquake Code (2007) and compared between 8 and
16 window openings cases. The results show that reduction of the window openings affected the structural behaviour of the mosque positively.

## 1 Introduction

Masonry is the oldest and a common construction technique in the world. It is a sustainable, easy and economical construction method but its share of the construction market has decreased in recent decades (Salmanpour et al., 2015). This
is because about new construction techniques. Even though many constructions technic have been developed and used in the last century, almost all historical structures all over the world made up with unreinforced masonry (Chisari et al., 2015). Masonry is a non-homogeneous material, formed by stone, brick, timber and mortar, which has different types, with distinct mechanical properties (Vasconcelos and Lourenço, 2009). Throughout the historical structures, stone material was used mostly for walls, bricks were used sometimes for wall and often for vaults, timber material was used generally for floor and
mortar was used as a bond material for walls. The construction of masonry structures which made from these materials are quite simple. Despite its simplicity of construction, the analysis and determining of the mechanical behaviour of masonry structures remains a challenge (Sarhosis et al., 2015). Their different building shapes, workmanships, materials and none homogeny construction types make them very abstruse to understand. When the structural behaviour of the masonry



structures examined it is seen that they have high compression strength because of stone and brick materials despite this their tensile strength is very low. Even if just a pinch mortar composes the tensile strength in masonry structures.

In the course of time masonry structures have been affected heavily by natural and man-made disasters. But one of the most destructive factor is earthquake. Earthquake creates horizontal load and this causes tensile stress at masonry structures which

have very low tensile strength. So it is of vital importance to investigate the earthquake behavior of the masonry structures. Earthquake behaviour of masonry structures are related to some parameters like material properties, shape of structure, support and load situations etc. Beside these an important factor there is exist in masonry structures namely openings in masonry walls. It is commonly accepted that the presence of openings reduces the lateral stiffness and strength of the infilled system (Chen and Liu, 2015). However it has still on conflict that how the openings are located in the systems and how

openings affect the earthquake behaviour of system. There are few studies made by researchers about this subject. In the Turkish Earthquake Code (2007) there is some information about openings in masonry but those are very limited (TEC, 2007).

There are many studies exist in the literature for masonry buildings like mosque, church, minaret and tower. Researchers investigated the masonry buildings different aspects but almost never studies exist in literature about openings in masonry.

Ranalli et al. (2004) investigated the preliminary stage of a project of structural monitoring and restoration of the facade of the Collemaggio Basilica (Italy) with ground penetrating radar (GPR) surveys which are non-destructive method. Mohebkhah et al. (2008) studied nonlinear analyses of masonry-infilled steel frames with openings using discrete element method. Shariq et al. (2008) investigated the influence of openings on seismic performance of masonry building walls. Bayraktar et al. (2009) emerged the modal parameter of Hagia Sophia Bell-Tower in terms of ambient vibration test.

Vasconcelos and Lourenço (2009) investigated the experimental characterization of stone masonry in shear and compression cases. Bosiljkov et al. (2010) assessed the historic masonry structures with an integrated diagnostic approach. Zimmermann et al. (2010) investigated the historic masonry walls under normal and shear loads. Altunişik (2011) determined the dynamic response of Iskenderpaşa masonry minaret strengthened with Fiber Reinforced Polymer (FRP) composites. Bayraktar et al. (2011) investigated the seismic behaviour of a minaret belong to Iskenderpaşa historical masonry mosque with a finite

element model which updated operational modal testing. Capozucca (2011) made experimental analysis on historic masonry walls which reinforced with carbon fiber reinforced polymers (CFRP). Erkal et al. (2012) examined wind-driven rain impact on surface erosion and surface strength reduction of historic building materials. Lin et al. (2012) investigated structural behaviour of the wall-diaphragm connections for older masonry buildings with experiments. Parisi et al. (2013) assessed rocking response of in-plane laterally-loaded masonry walls with openings. Betti et al. (2014) compared different methods of

analysis and different numerical models with through the investigation of a reference masonry prototype, for to estimate the seismic behaviour of unreinforced masonry buildings. Mazzotti et al. (2014) investigated the shear strength of historic masonries with moderately destructive testing of masonry cores. Chen and Liu (2015) investigated the in-plane behaviour and strength of concrete masonry infills with openings. Sarhosis et al. (2015), examined the influence of the brick–mortar interface on the pre and post-cracking behaviour of low bond strength masonry wall panels under vertical loads.





Mosques which have beautiful appearance and high religious value are very important for Muslims. Many mosques have been built in Muslims area and most of them were built with masonry technics. There are lots of masonry mosques in Turkey and most of these mosques are especially on seismic zones. The presence of earthquakes in Turkey caused damage and destructions on masonry mosques. Also Hüsrev Pasha Mosque which was located in active seismic zone was damaged from

some seismic events. In the restoration of Hüsrev Pasha Mosque there is being applied an application which is about reduction of window openings on dome for to improve the structural performance. So it is required to investigate this case. For this purpose the restoration effect also reduction of window openings effect on the earthquake response of masonry mosques are investigated in this paper.

## 2 Hüsrev Pasha Mosque

Hüsrev Pasha Mosque is located in Ortakapı district in old city of Van, Turkey and it is in the central of Hüsrev Pasha Islamic-Ottoman social complex. The mosque was built by Köse Hüsrev Mehmed Pasha which was governor of Van and vizier of Süleyman the magnificent. The mosque had been constructed by famous Turkish architect Mimar Sinan between 1567 and 1568.

The mosque has one big and five small domes. The big ones and the smalls covered mean prayer area and last prayer area
respectively. Mean prayer area has 15.00m x 15.20m geometric shape and the walls which surround this area is 2m. The walls consist of cut stone and rough stones, transition elements and domes consist of bricks material. Limestone was used for bonding in the walls. The mosque has a minaret which has square shape base and cylindrical body. There were china at the walls up to 2m height and pencil arts on the dome in the mosque but most of them haven't reached today. Exterior parts of the mosque have colorful stonemasonry namely red-white stones up to underside of windows and black-white stones rest of
the walls with red ribbon. There is an historical inscription above the mosque's door.

The mosque is located at an active seismic region and classified as second degree earthquake zone in the Seismic Zoning Map which was published by the Ministry of Public Works and Settlement of Turkey in 1996. Earthquake map of Turkey and Van are shown in Fig. 1a and Fig. 1b (AFAD, 2015). Because of this characteristic the mosque was damaged severely by earthquakes. The mosque and social complex which had served for centuries were damaged by 1839 earthquake and fire.
After this sadness event the complex was renewed. During the World War I the complex and mosque were vandalised heavily. Beside these events the complex and mosque were ruined by 2011 Van Earthquakes. The minaret and last prayer area were heavily damaged in these earthquakes. Today the prayer area and shrine are healthy but rest of the complex highly damaged or destroyed. For this reason the mosque was closed for praying and it is aimed to open the mosque for praying with restoration project. Undamaged views of the mosque are given in Fig. 2.



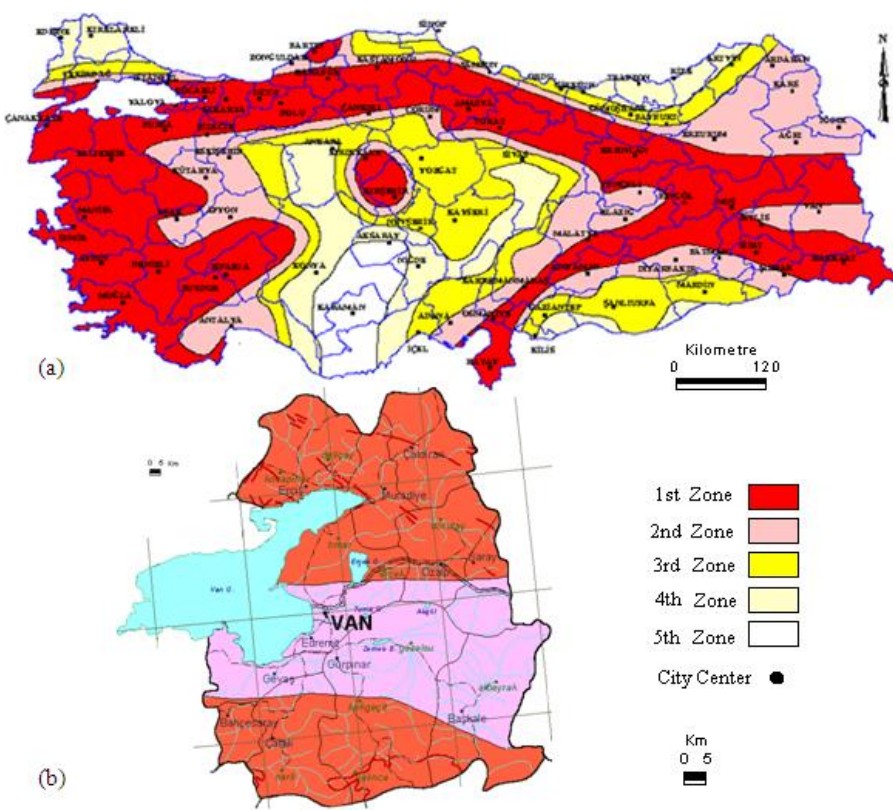

**Figure 1: Seismic zoning map of Turkey (a) and Van (b) (AFAD, 2015).**

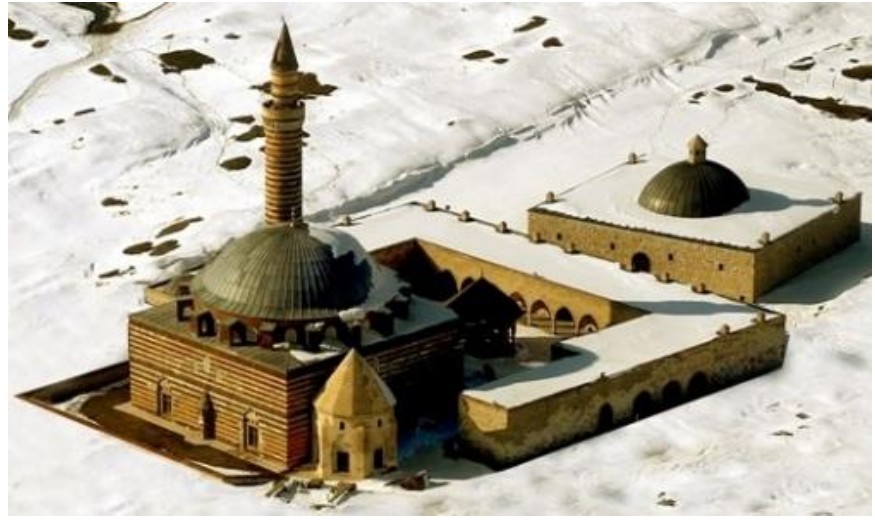

**Figure 2: Undamaged view of Hüsrev Pasha Mosque and social complex (URL-1).**





Today there are some cases which affect the structural performance of the mosque detrimentally. These cases are structural cracks, material deformations, destroyed parts of carrier system, environmental issues-algae and humidity. Some views of these cases and damaged views of the mosque are shown in Fig. 3.

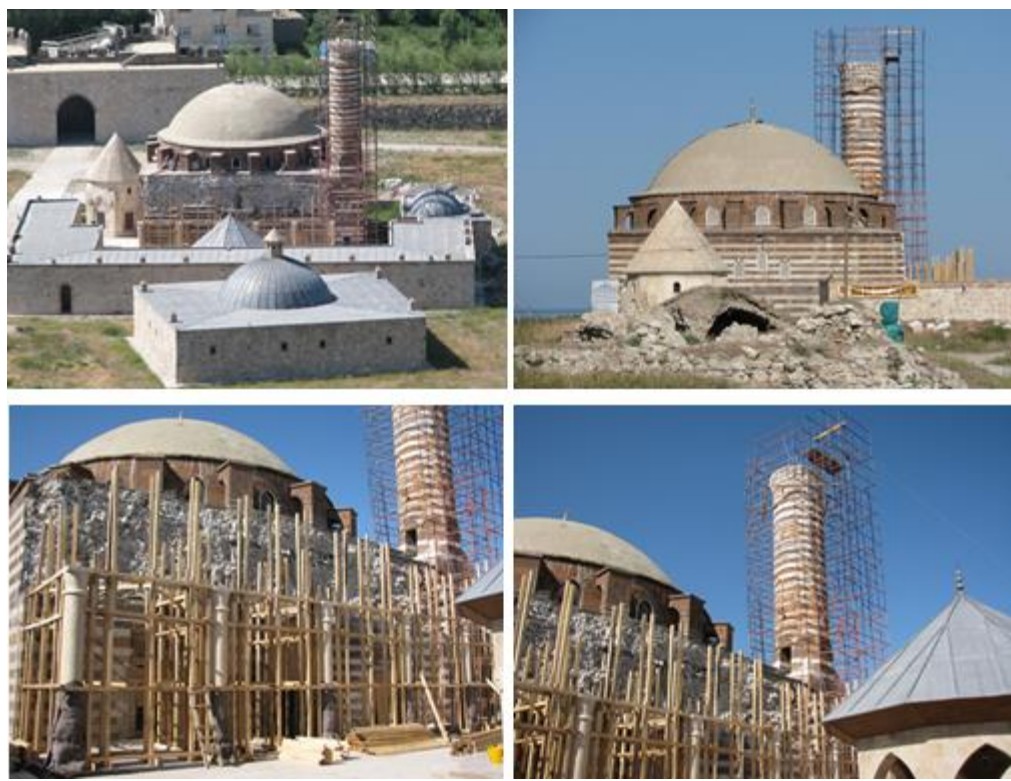

5  **Figure 3: Damaged views and dangerous cases of Hüsrev Pasha Mosque.**

## 3 Structural Analysis of the Mosque

Finite element analyses were made for investigate the restoration effect on the earthquake behaviour of the mosque considering different opening ratios on dome. The model was constituted with restoration drawings considering 8 and 16 openings on dome. Required information for modelling and analysing obtained from contractor firm, restoration drawings

10  and other sources. Finite element models of the mosque were created with SAP2000 structural analysis program (SAP2000, 2015). Earthquake analyses were affected x, y, z directions for better comprehend the restoration effect with different openings on dome. After analyses maximum displacements, maximum-minimum principal stresses and shear stresses were obtained and results are given with contour diagrams.

Structural analyses of the mosque are listed below:

Structural analyses of the mosque with 16 window openings





- *Modal analysis of the mosque*

- *Dead load and earthquake load (x direction)*

- *Dead load and earthquake load ( y direction)*

- *Dead load and earthquake load ( z direction)*

Structural analyses of the mosque with 8 window openings

- *Modal analysis of the mosque*

- *Dead load and earthquake load (x direction)*

- *Dead load and earthquake load ( y direction)*

- *Dead load and earthquake load ( z direction)*

Finite element models of the mosque were constituted with using bar, area and 3D solid elements in SAP2000 program. The bar, area and 3D solid elements have 2, 4 and 12 nodes respectively and each node has three degrees of freedom.

Linear elastic methods cannot give exact solutions in the structural analyses of historical masonry structures which are built with brick and stone masonry. Despite that, nonlinear analyses can give exact results if material properties of masonry are defined in analyses correctly. But if the analysed structures are very big and complex there could be some problems at

iterations in nonlinear analyses. Also defining the material properties of masonry structures are fairly difficult because of their non-homogeneity. For these reasons all analyses were made as linear elastic. Material properties which were used in the analyses are given with Table 1.

**Table 1 Material properties of Hüsrev Pasha Mosque**

| Carrier System Components | Material Properties | | |
|---|---|---|---|
| | *Modulus of Elasticity* $(N/m^2)$ | *Poisson Ratio* (-) | *Density* $(kg/m^3)$ |
| *Prayer Location* | | | |
| Cut Stones | 1.60E09 | 0.200 | 2000 |
| Artless Stone (*with considering the mortar*) | 4.50E08 | 0.200 | 2400 |
| Brick | 1.20E09 | 0.200 | 2400 |
| *Outer Part of Prayer Location* | | | |
| Cut Stones | 1.60E09 | 0.200 | 2000 |
| Marble | 3.54E10 | 0.316 | 2690 |
| Stretcher | 2.00E11 | 0.300 | 7850 |





Finite element analyses results; displacement, section effect and stress values were controlled with considering the allowed values in 2007 Turkish Earthquake Code (TEC, 2007)-Chapter 5.3(Pressure Safety Stresses of Walls in which Free Pressure Strength is Unknown). Safety stresses for Husrev Pasha Mosque are given below in Table 2.

**Table 2 Safety stresses of materials.**

| Materials | Material Properties | | |
|---|---|---|---|
| | *Pressure Safety stresses (MPa)* | *Tension Safety stresses (MPa)* | *Shear Safety stresses (MPa)* |
| Brick | 2.40 | 0.360 | 1.05 |
| Stone wall | 0.90 | 0.135 | 0.53 |
| Arches | 0.90 | 0.135 | 0.53 |

Mode Superposition Method was used in earthquake analyses and elastic behaviour of the mosque under vertical and earthquake loads were obtained with using the full square consolidation method. Spectral acceleration coefficient was taken as $S(T)=2.5$ and effective earthquake coefficient was chosen as $A_o=0.3$ for related the region which is in second seismic

zone. In masonry structures when the tension stresses reached the tension strength of material, cracks and weakness occur. This situation is taken into consideration in the analyses with using the earthquake load reduction coefficient ($R_a$) in each period. There is no reduction made in this study ($R_a=1$) but safety stresses are enlarged with 3.

Hüsrev Pasha Mosque wasn't built with considering any building code; it was also built with experience. There are some rules about openings for masonry in the Turkish Earthquake Code (2007). When the mosque is assessed with considering

these rules, there has no discrepancy with code. Some rules which are exist in the code about openings for masonry related to this study given below (TEC, 2007):

   • Plan length of the solid masonry wall which is between the corner of a building and the nearest window or door opening shall not be less than 1.50 m for the first and second seismic zones and 1.0 m for the third and fourth seismic zones.

• Plan length of the solid masonry wall which is between window and door openings shall not be less than 1.0 m for the first and second seismic zones and 0.8 m for the third and fourth seismic zones.

   • Except for the corners of buildings, plan length of a solid masonry wall which is between intersection of the walls and the nearest window or door opening to the intersection of the orthogonal walls shall not be less than 0.50 m in the all seismic zones.

• Plan length of each window or door opening shall not be more than 3.0 m.



## 3.1 Structural analyses of the mosque with 16 window openings

The mosque has 16 window openings before the restoration so finite element model of the mosque was constituted with 16 window openings. The model of the mosque has 27297 nodes, 127 bar elements, 25653 area elements and 102460 3D solid elements. Finite element model of the mosque with 16 window openings is shown in Fig. 4.

**Figure 4: Finite element model of the mosque with 16 window openings.**



### 3.1.1 Defining the dynamic characteristics and modal analysis of the mosque with 16 window openings

Dynamic characteristics were obtained with modal analysis. %5 damping ratio was used in the analysis. 20 mode shapes were obtained after analyses. First four mode shape and frequency are given with Fig. 5.

1. Mode ($f_1$=3.74 Hz)
Translation (X direction)

2. Mode ($f_2$=3.78 Hz)
Translation (Y direction)

3. Mode ($f_3$=4.73 Hz)
Squeeze

4. Mode ($f_4$=5.66 Hz)
Torsion

5  **Figure 5: First four mode shape and frequency.**




### 3.1.2 Structural response of the mosque with 16 window openings under dead load and horizontal earthquake load (G+EX)

The maximum displacements contour diagram of Hüsrev Pasha Mosque with 16 window openings under dead load and earthquake load (G+EX) is shown in Fig. 6. It can be seen from the Figure 6 that the maximum displacement occurred at the middle point of the big dome as 42.0 mm. Beside this displacements have a decreasing trend from top of the dome to lower

5   part of the mosque.

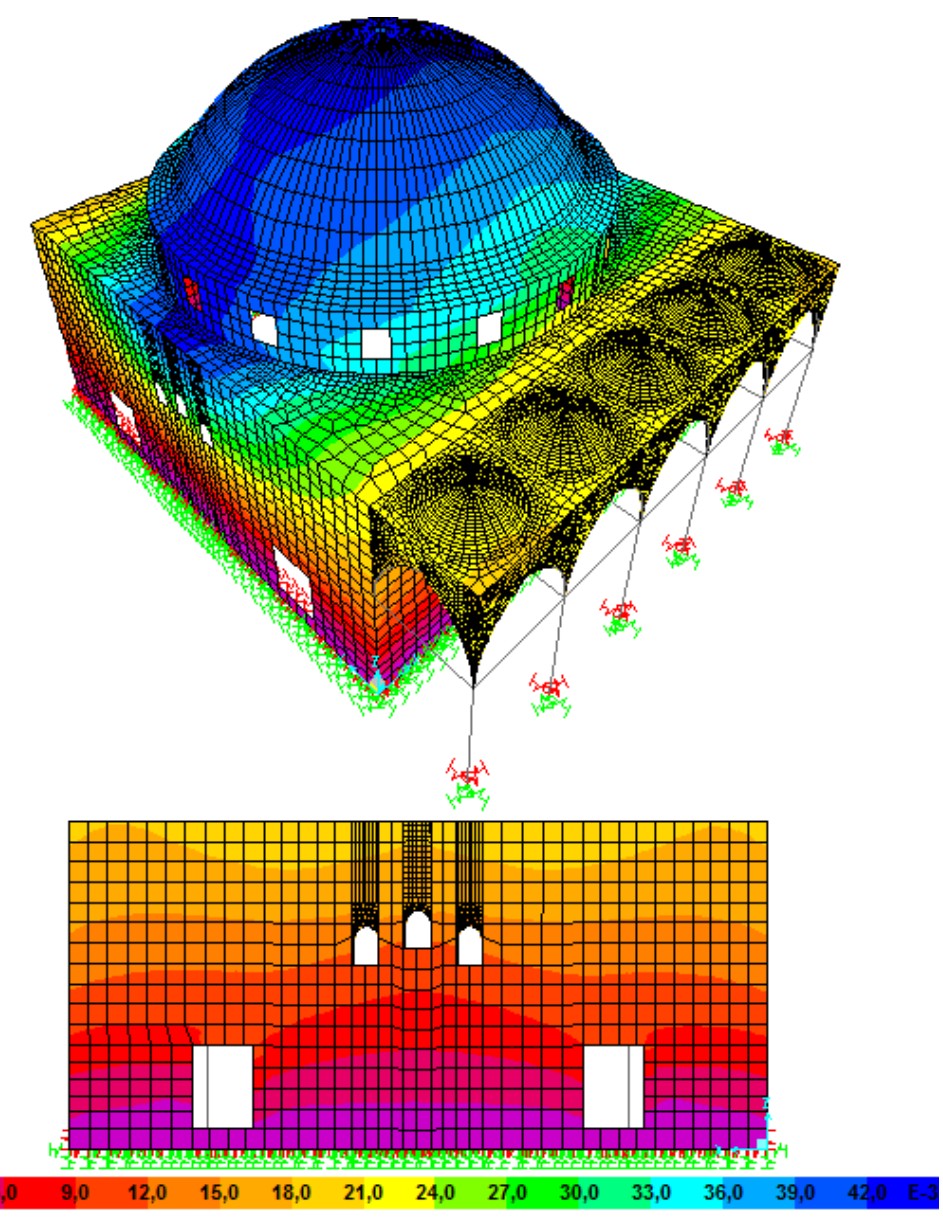

**Figure 6: Maximum displacements contour diagram of the mosque with 16 window openings under dead load and earthquake load (G+EX).**





The maximum tensile stresses contour diagram for outer and inner surfaces of Hüsrev Pasha Mosque with 16 window openings under dead load and earthquake load (G+EX) is shown in Fig. 7. It is seen from the Figure 7 that maximum values of the tensile stresses for outer surface of the mosque occurred at side and lower part of big dome, near the window spaces and transition areas of between side walls as 1.05 MPa. Maximum tensile stresses for inner surface of the mosque occurred

5   as 1.00 MPa.

**Figure 7: Maximum tensile stresses contour diagram for outer and inner surfaces of the mosque with 16 window openings under dead load and earthquake load (G+EX).**





The maximum compression stresses contour diagram for outer and inner surfaces of Hüsrev Pasha Mosque with 16 window openings under dead load and earthquake load (G+EX) is shown in Fig. 8. It is seen from the Figure 8 that maximum values of the compression stresses for outer surface of mosque occurred at between the dome and side walls transition areas and near window spaces at dome as 2.05 MPa. Beside this maximum compression stresses for inner surface of the mosque,

5      occurred as 1.85 MPa.

**Figure 8: Maximum compression stresses contour diagram for outer and inner surfaces of the mosque with 16 window openings under dead load and earthquake load (G+EX).**



The maximum shear stresses contour diagram for outer and inner surfaces of Hüsrev Pasha Mosque under dead load and earthquake load (G+EX) is shown in Fig. 9. It is seen from the Figure 9 that maximum values of the shear stresses for outer and inner surfaces of the mosque are 0.65 MPa and 0.60 MPa respectively.

**Figure 9: Maximum shear stresses contour diagram for outer and inner surfaces of the mosque with 16 window openings under dead load and earthquake load (G+EX).**



### 3.1.3 Structural response of the mosque with 16 window openings under dead load and horizontal earthquake load (G+EY)

The maximum displacements contour diagram of Hüsrev Pasha Mosque with 16 window openings under dead load and earthquake load (G+EY) is shown in Fig. 10. It can be seen from the Figure 10 that the maximum displacement occurred at the middle point of the big dome as 44.0 mm. Beside this displacements have a decreasing trend from top of the dome to lower part of the mosque.

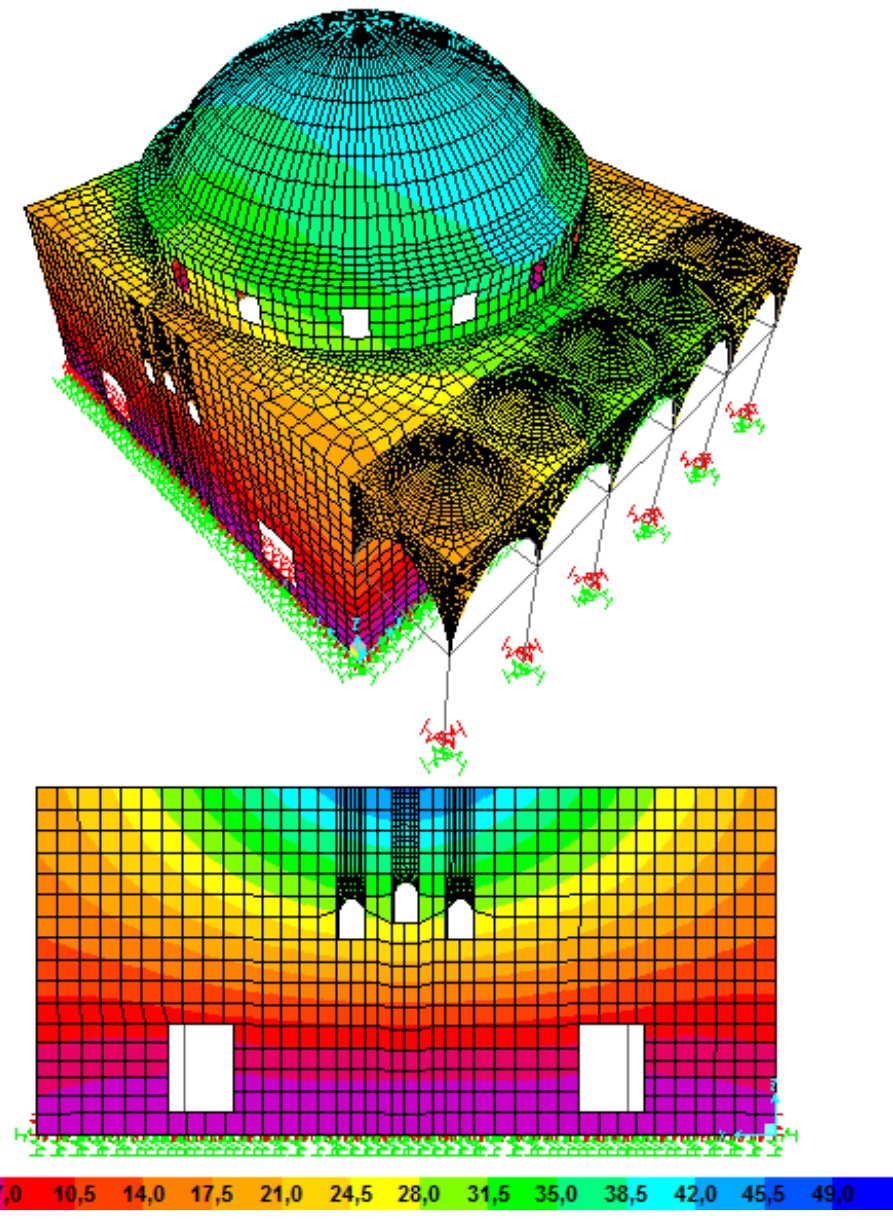

**Figure 10: Maximum displacements contour diagram of the mosque with 16 window openings under dead load and earthquake load (G+EY).**



The maximum tensile stresses contour diagram for outer and inner surfaces of Hüsrev Pasha Mosque with 16 window openings under dead load and earthquake load (G+EY) is shown in Fig. 11. It is seen from the Figure 11 that maximum values of the tensile stresses for outer surface of the mosque occurred at side and lower part of big dome, near the window spaces and transition areas of between side walls as 0.95 MPa. Maximum tensile stresses for inner surface of the mosque

5   occurred as 0.75 MPa.

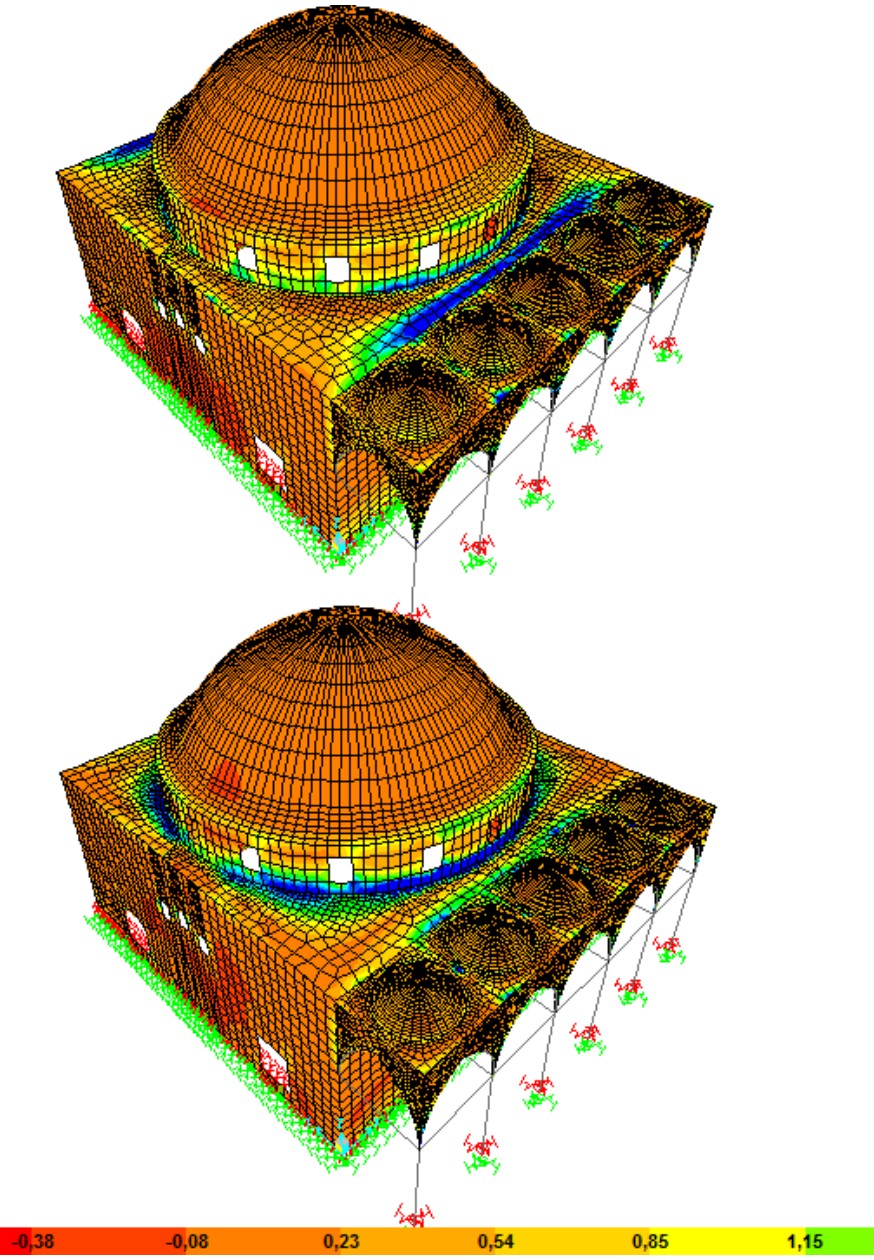

**Figure 11: Maximum tensile stresses contour diagram for outer and inner surfaces of the mosque with 16 window openings under dead load and earthquake load (G+EY).**




The maximum compression stresses contour diagram for outer and inner surfaces of Hüsrev Pasha Mosque with 16 window openings under dead load and earthquake load (G+EY) is shown in Fig. 12. It is seen from the Figure 12 that maximum values of the compression stresses for outer surface of mosque occurred at between the dome and side walls transition areas and near window spaces at dome as 1.15 MPa. Beside this maximum compression stresses for inner surface of the mosque, occurred as 1.10 MPa.

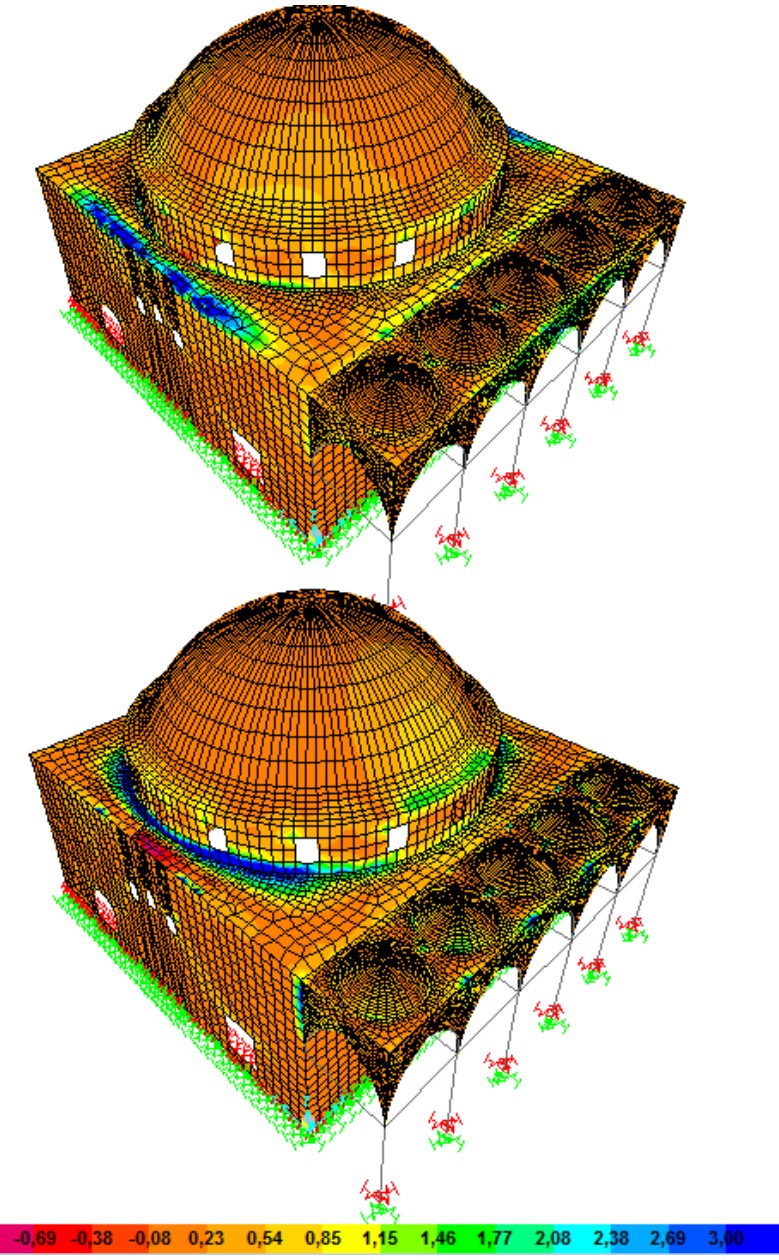

**Figure 12: Maximum compression stresses contour diagram for outer and inner surfaces of the mosque with 16 window openings under dead load and earthquake load(G+EY).**



The maximum shear stresses contour diagram for outer and inner surfaces of Hüsrev Pasha Mosque under dead load and earthquake load (G+EY) is shown in Fig. 13. It is seen from the Figure 13 that maximum values of the shear stresses for outer and inner surfaces of the mosque are 0.60 MPa and 0.55 MPa respectively.

5   **Figure 13: Maximum shear stresses contour diagram for outer and inner surfaces of the mosque with 16 window openings under dead load and earthquake load (G+EY).**



### 3.1.4 Structural response of the mosque with 16 window openings under dead load and horizontal earthquake load (G+EZ)

The maximum displacements contour diagram of Hüsrev Pasha Mosque with 16 window openings under dead load and earthquake load (G+EZ) is shown in Fig. 14. It can be seen from the Figure 14 that the maximum displacement occurred at

5   the middle point of the big dome as 19.6 mm. Beside this displacements have a decreasing trend from top of the dome to lower part of the mosque.

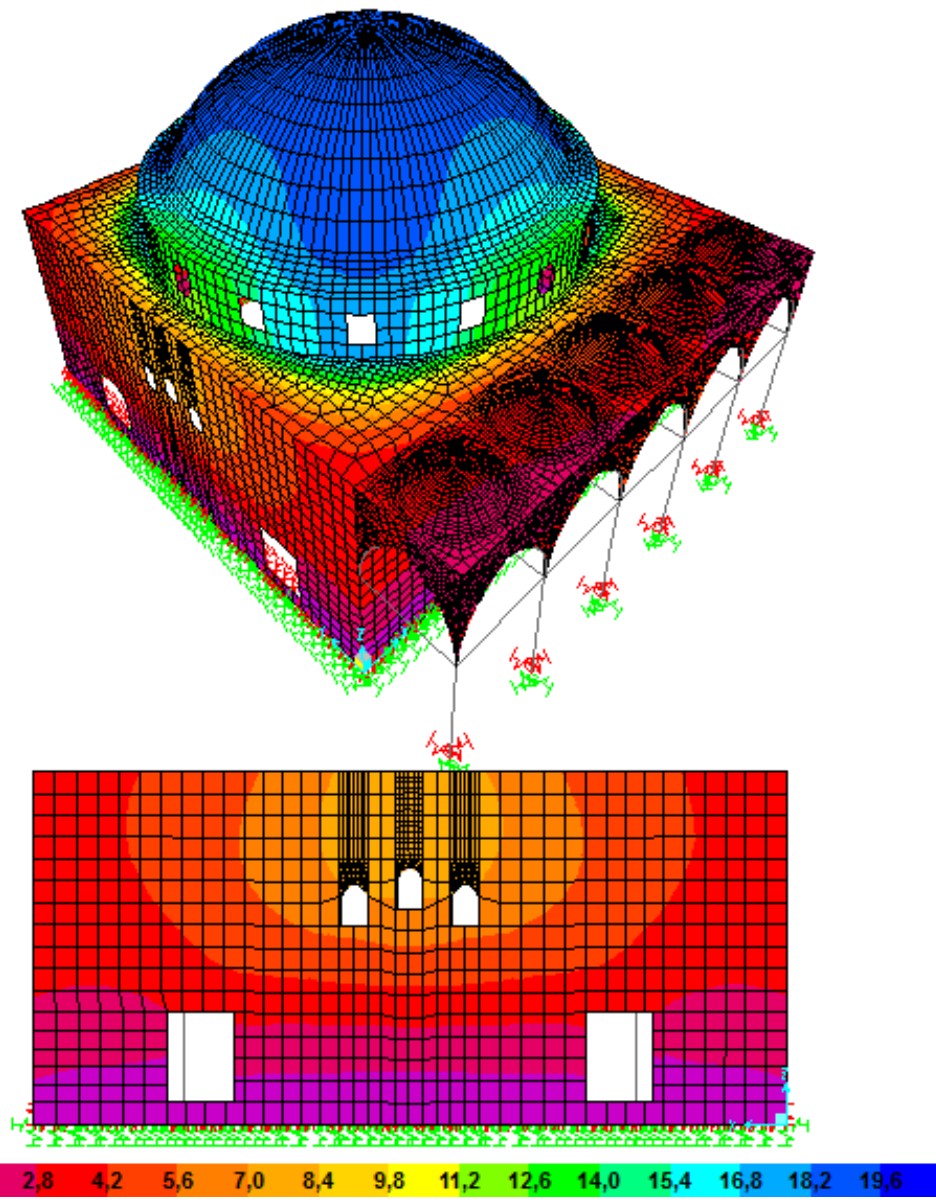

**Figure 14: Maximum displacements contour diagram of the mosque with 16 window openings under dead load and earthquake load (G+EZ).**





The maximum tensile stresses contour diagram for outer and inner surfaces of Hüsrev Pasha Mosque with 16 window openings under dead load and earthquake load (G+EZ) is shown in Fig. 15. It is seen from the Figure 15 that maximum values of the tensile stresses for outer surface of the mosque occurred at side and lower part of big dome, near the window spaces and transition areas of between side walls as 0.85 MPa. Maximum tensile stresses for inner surface of the mosque

5   occurred as 0.80 MPa.

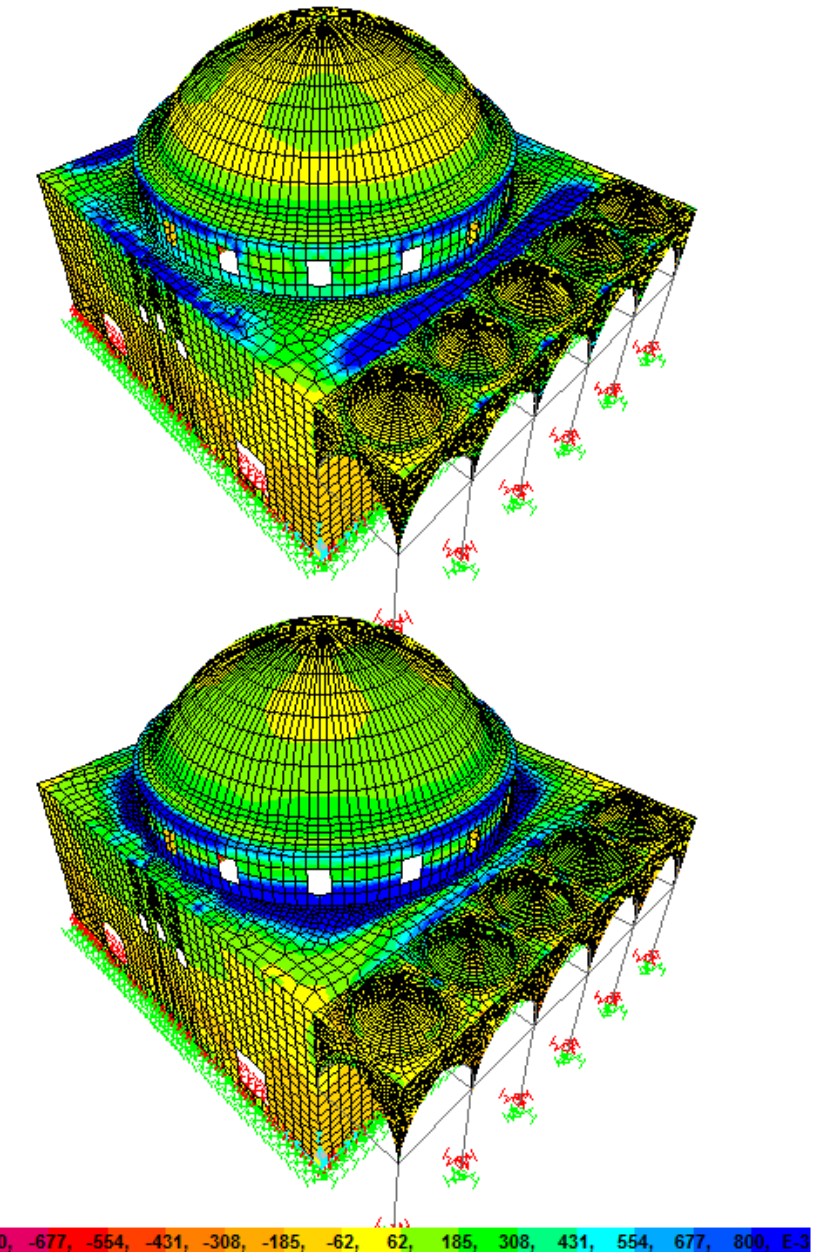

**Figure 15: Maximum tensile stresses contour diagram for outer and inner surfaces of the mosque with 16 window openings under dead load and earthquake load (G+EZ).**





The maximum compression stresses contour diagram for outer and inner surfaces of Hüsrev Pasha Mosque with 16 window openings under dead load and earthquake load (G+EZ) is shown in Fig. 16. It is seen from the Figure 16 that maximum values of the compression stresses for outer surface of mosque occurred at between the dome and side walls transition areas and near window spaces at dome as 1.35 MPa. Beside this maximum compression stresses for inner surface of the mosque, occurred as 1.25 MPa.

**Figure 16: Maximum compression stresses contour diagram for outer and inner surfaces of the mosque with 16 window openings under dead load and earthquake load (G+EZ).**



The maximum shear stresses contour diagram for outer and inner surfaces of Hüsrev Pasha Mosque under dead load and earthquake load (G+EZ) is shown in Fig. 17. It is seen from the Figure 17 that maximum values of the shear stresses for outer and inner surfaces of the mosque are 0.50 MPa and 0.45 MPa respectively.

**Figure 17: Maximum shear stresses contour diagram for outer and inner surfaces of the mosque with 16 window openings under dead load and earthquake load (G+EZ).**





## 3.2 Structural analyses of the mosque with 8 window openings

The mosque has 8 window openings before the restoration case so finite element model of the mosque was constituted with 8 window openings. Finite element model of the mosque has 27305 nodes, 127 bar elements, 25685 area elements and 102588 3D solid elements. Finite element model of the mosque with 8 window openings is shown in Fig. 18.

**Figure 18: Finite element model of the mosque with 8 window openings.**



### 3.2.1 Defining the dynamic characteristics and modal analysis of the mosque with 8 window openings

Dynamic characteristics were obtained with modal analysis. %5 damping ratio was used in the analysis. 20 mode shapes were obtained after analyses. First four mode shape and frequency are given with Fig. 19.

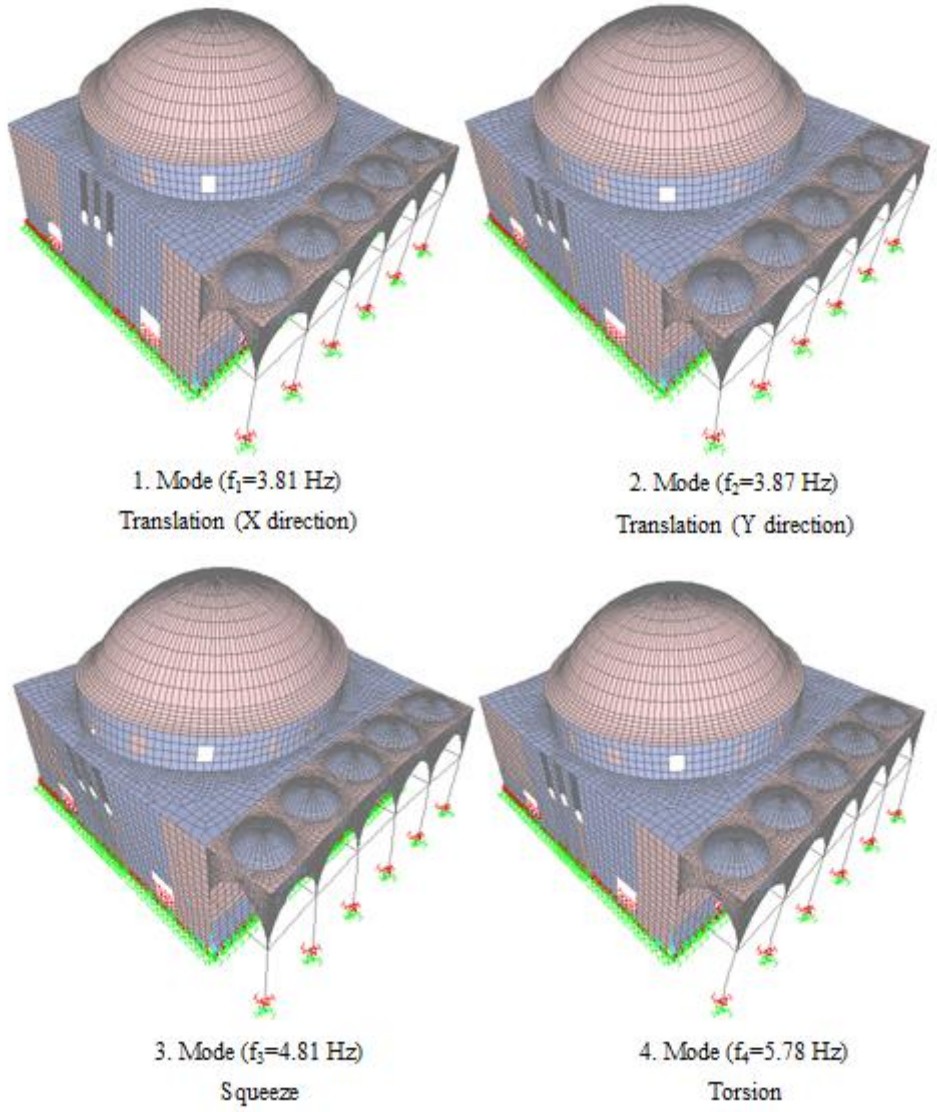

1. Mode ($f_1$=3.81 Hz)
Translation (X direction)

2. Mode ($f_2$=3.87 Hz)
Translation (Y direction)

3. Mode ($f_3$=4.81 Hz)
Squeeze

4. Mode ($f_4$=5.78 Hz)
Torsion

5   **Figure 19: First four mode shape and frequency.**


### 3.2.2 Structural response of the mosque with 8 window openings under dead load and horizontal earthquake load (G+EX)

The maximum displacements contour diagram of Hüsrev Pasha Mosque with 8 window openings under dead load and earthquake load (G+EX) is shown in Fig. 20. It can be seen from the Figure 20 that the maximum displacement occurred at

5    the middle point of the big dome as 36.0mm. Beside this displacements have a decreasing trend from top of the dome to lower part of the mosque.

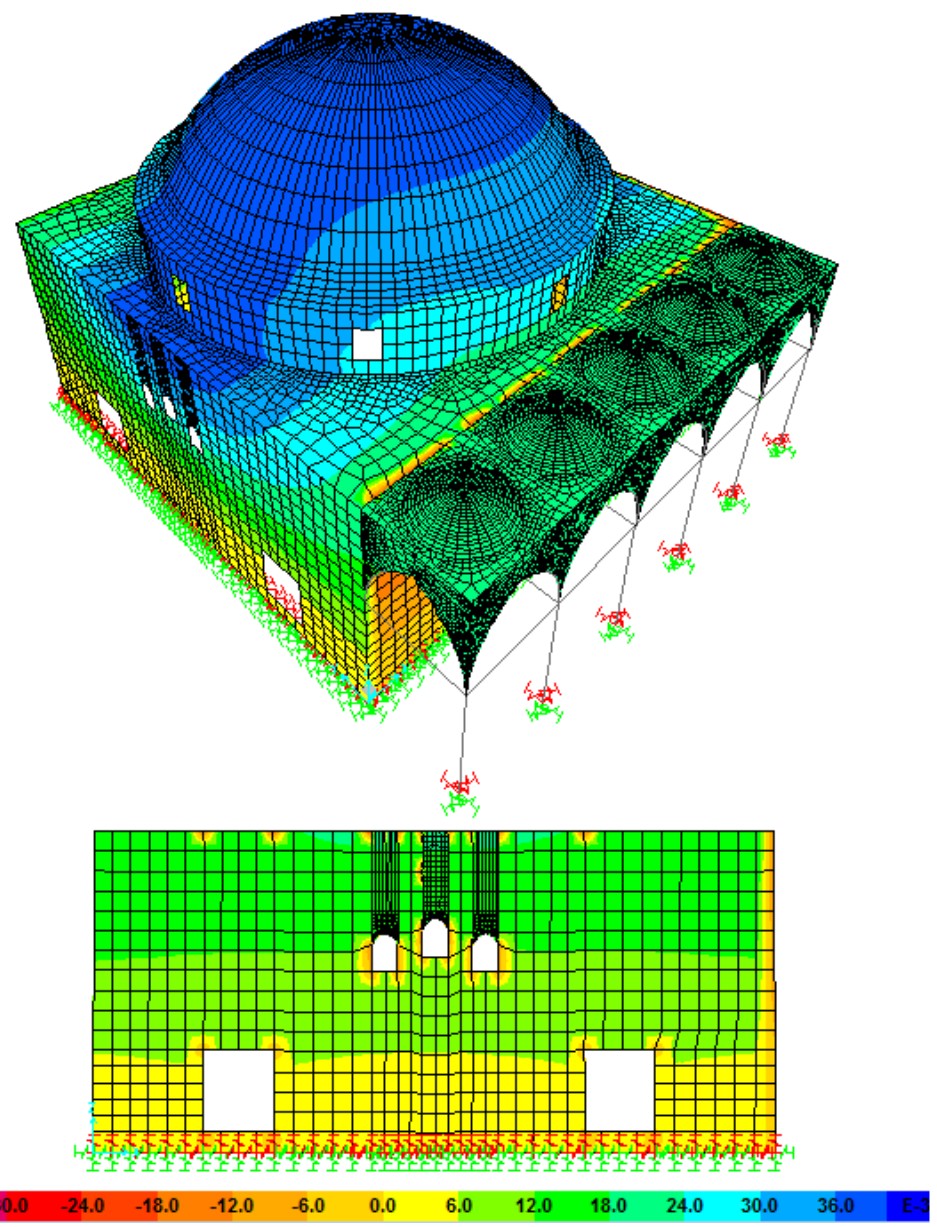

**Figure 20: Maximum displacements contour diagram of the mosque with 8 window openings under dead load and earthquake load (G+EX).**





The maximum tensile stresses contour diagram for outer and inner surfaces of Hüsrev Pasha Mosque with 8 window openings under dead load and earthquake load (G+EX) is shown in Fig. 21. It is seen from the Figure 21 that maximum values of the tensile stresses for outer surface of the mosque occurred at side and lower part of big dome, near the window spaces and transition areas of between side walls as 0.95 MPa. Maximum tensile stresses for inner surface of the mosque

5    occurred as 0.90 MPa.

**Figure 21: Maximum tensile stresses contour diagram for outer and inner surfaces of the mosque with 8 window openings under dead load and earthquake load (G+EX).**


The maximum compression stresses contour diagram for outer and inner surfaces of Hüsrev Pasha Mosque with 8 window openings under dead load and earthquake load (G+EX) is shown in Fig. 22. It is seen from the Figure 22 that maximum values of the compression stresses for outer surface of mosque occurred at between the dome and side walls transition areas and near window spaces at dome as 1.74 MPa. Beside this maximum compression stresses for inner surface of the mosque,

5    occurred as 1.51 MPa.

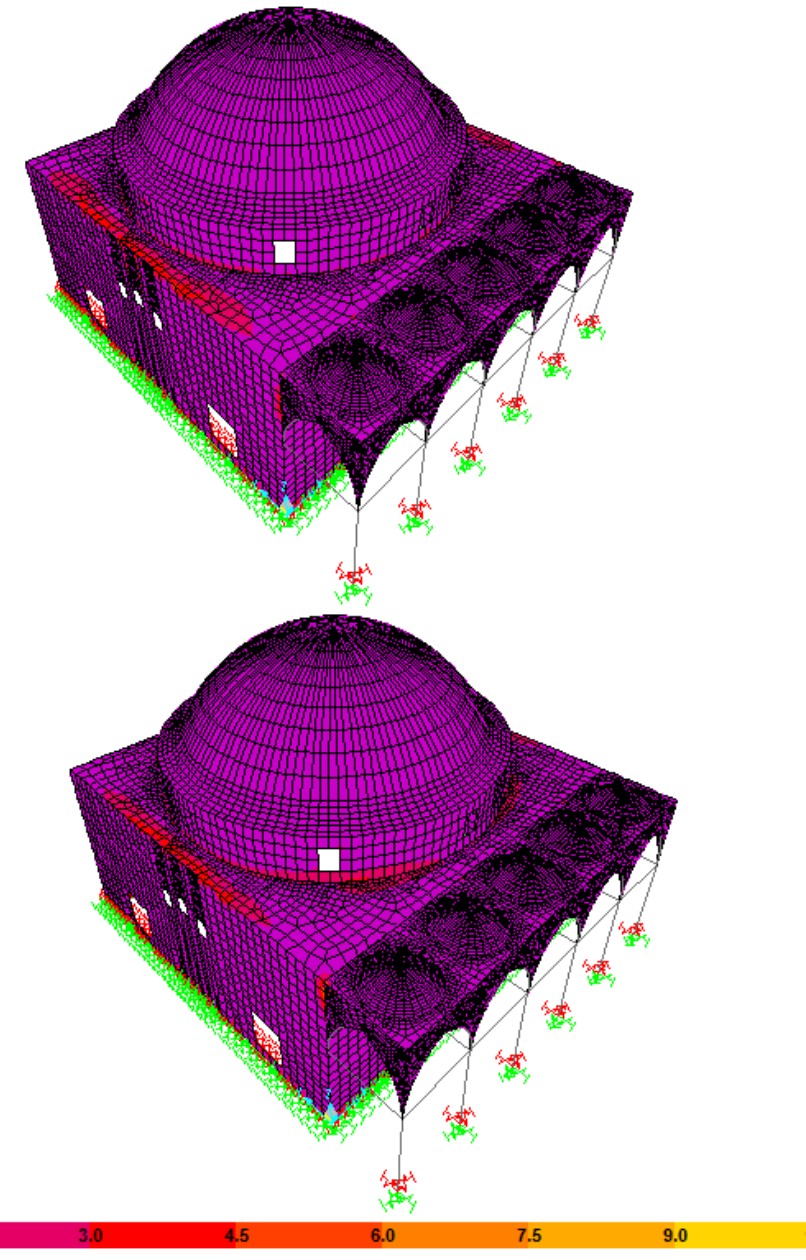

**Figure 22: Maximum compression stresses contour diagram for outer and inner surfaces of the mosque with 8 window openings under dead load and earthquake load (G+EX).**





The maximum shear stresses contour diagram for outer and inner surfaces of Hüsrev Pasha Mosque with 8 window openings under dead load and earthquake load (G+EX) is shown in Fig. 23. It is seen from the Figure 23 that maximum values of the shear stresses for outer and inner surfaces of the mosque are 0.55 MPa and 0.50 MPa respectively.

5 **Figure 23: Maximum shear stresses contour diagram for outer and inner surfaces of the mosque with 8 window openings under dead load and earthquake load (G+EX).**





### 3.2.3 Structural response of the mosque with 8 window openings under dead load and horizontal earthquake load (G+EY)

The maximum displacements contour diagram of Hüsrev Pasha mosque with 8 window openings under dead load and earthquake load (G+EY) is shown in Fig. 24. It can be seen from the Figure 24 that the maximum displacement occurred at

5  the middle point of the big dome as 39.0 mm. Beside this displacements have a decreasing trend from top of the dome to lower part of the mosque.

**Figure 24: Maximum displacements contour diagram of the mosque with 8 window openings under dead load and earthquake load (G+EY).**


The maximum tensile stresses contour diagram for outer and inner surfaces of Hüsrev Pasha Mosque with 8 window openings under dead load and earthquake load (G+EY) is shown in Fig. 25. It is seen from the Figure 25 that maximum values of the tensile stresses for outer surface of the mosque occurred at side and lower part of big dome, near the window spaces and transition areas of between side walls as 0.80 MPa. Maximum tensile stresses for inner surface of the mosque

5 occurred as 0.65 MPa.

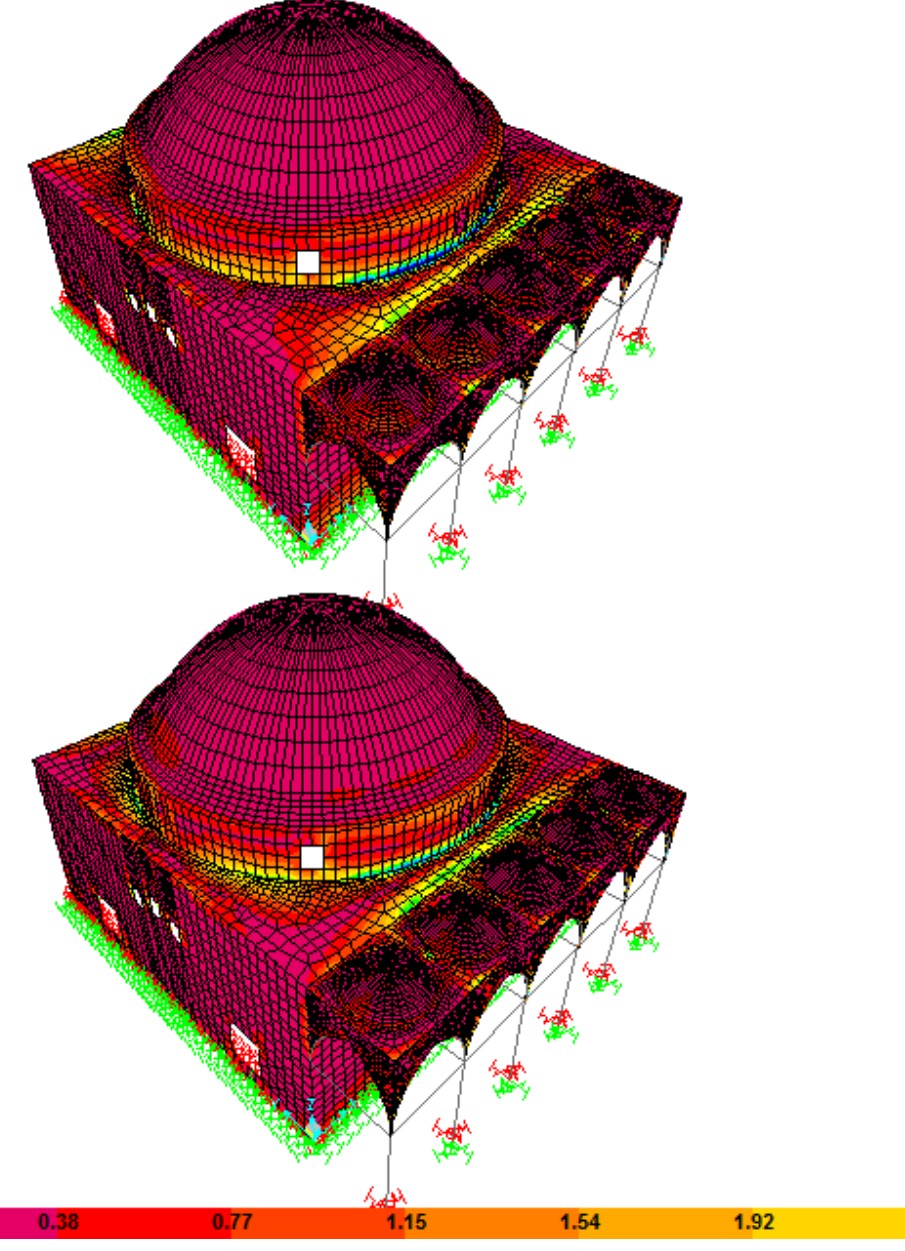

**Figure 25: Maximum tensile stresses contour diagram for outer and inner surfaces of the mosque with 8 window openings under dead load and earthquake load (G+EY).**



The maximum compression stresses contour diagram for outer and inner surfaces of Hüsrev Pasha Mosque with 8 window openings under dead load and earthquake load (G+EY) is shown in Fig. 26. It is seen from the Figure 26 that maximum values of the compression stresses for outer surface of mosque occurred at between the dome and side walls transition areas and near window spaces at dome as 1.00 MPa. Beside this maximum compression stresses for inner surface of the mosque, occurred as 0.95 MPa.

**Figure 26: Maximum compression stresses contour diagram for outer and inner surfaces of the mosque with 8 window openings under dead load and earthquake load (G+EY).**



The maximum shear stresses contour diagram for outer and inner surfaces of Hüsrev Pasha Mosque with 8 window openings under dead load and earthquake load (G+EY) is shown in Fig. 27. It is seen from the Figure 27 that maximum values of the shear stresses for outer and inner surfaces of the mosque are 0.50 MPa and 0.45 MPa respectively.

**Figure 27: Maximum shear stresses contour diagram for outer and inner surfaces of the mosque with 8 window openings under dead load and earthquake load (G+EY).**





### 3.2.4 Structural response of the mosque with 8 window openings under dead load and horizontal earthquake load (G+EZ)

The maximum displacements contour diagram of Hüsrev Pasha Mosque with 8 window openings under dead load and earthquake load (G+EZ) is shown in Fig. 28. It can be seen from the Figure 28 that the maximum displacement occurred at

5   the middle point of the big dome as 9.80 mm. Beside this displacements have a decreasing trend from top of the dome to lower part of the mosque.

**Figure 28: Maximum displacements contour diagram of the mosque with 8 window openings under dead load and earthquake load (G+EZ).**



The maximum tensile stresses contour diagram for outer and inner surfaces of Hüsrev Pasha Mosque with 8 window openings under dead load and earthquake load (G+EZ) is shown in Fig. 29. It is seen from the Figure 29 that maximum values of the tensile stresses for outer surface of the mosque occurred at side and lower part of big dome, near the window spaces and transition areas of between side walls as 0.69 MPa. Maximum tensile stresses for inner surface of the mosque

5  occurred as 0.58 MPa.

**Figure 29: Maximum tensile stresses contour diagram for outer and inner surfaces of the mosque with 8 window openings under dead load and earthquake load (G+EZ).**



The maximum compression stresses contour diagram for outer and inner surfaces of Hüsrev Pasha Mosque with 8 window openings under dead load and earthquake load (G+EZ) is shown in Fig. 30. It is seen from the Figure 30 that maximum values of the compression stresses for outer surface of mosque occurred at between the dome and side walls transition areas and near window spaces at dome as 1.15 MPa. Beside this maximum compression stresses for inner surface of the mosque,

5   occurred as 1.05 MPa.

**Figure 30: Maximum compression stresses contour diagram for outer and inner surfaces of the mosque with 8 window openings under dead load and earthquake load (G+EZ).**



The maximum shear stresses contour diagram for outer and inner surfaces of Hüsrev Pasha Mosque with 8 window openings under dead load and earthquake load (G+EZ) is shown in Fig. 31. It is seen from the Figure 31that maximum values of the shear stresses for outer and inner surfaces of the mosque are 0.30 MPa and 0.20 MPa respectively.

5    **Figure 31: Maximum shear stresses contour diagram for outer and inner surfaces of the mosque with 8 window openings under dead load and earthquake load (G+EZ).**





After the analyses, maximum displacements, maximum compression and tension stresses and maximum shear stresses results for 16 window openings and 8 window openings cases collected in Table 3 and Table 4 respectively.

According to the results the displacements reduce related to reduction of window openings. These reduction ratios are %14, %11 and %50 for dead and earthquake analyses with x, y and z directions respectively. This result shows that reduction of
window openings affects the displacements substantially.

The results show that the compression stresses decrease when the window openings are reduced. At the outer side of the mosque these reduction ratios are %15, %13 and %15 for dead and earthquake analyses with x, y and z directions respectively. Beside this at the inner side of the mosque the reduction ratios are %18, %13 and %16 for dead and earthquake analyses with x, y and z directions respectively.

The results show that the tension stresses decrease when the window openings are reduced. At the outer side of the mosque these reduction ratios are %9, %16 and %19 for dead and earthquake analyses with x, y and z directions respectively. Beside this at the inner side of the mosque the reduction ratios are %10, %13 and %27 for dead and earthquake analyses with x, y and z directions respectively.

It is seen from the tables the shear stresses decrease with the reduction of window openings. At the outer side of the mosque
these reduction ratios are %15, %16 and %40 for dead and earthquake analyses with x, y and z directions respectively. Beside this at the inner side of the mosque the reduction ratios are %16, %18 and %55 for dead and earthquake analyses with x, y and z directions respectively.

Stresses generally occur at near the openings. Therefore openings decrease the structure stability. With the reducing the opening ratios so ensure the integrity on the walls, the stresses and the displacements decrease substantially.

**Table 3 Whole analyses results of the mosque with 16 window openings case**

| Analyses Data | | Analyses | | |
| --- | --- | --- | --- | --- |
| | | *Dead and earthquake loads(X direction)* | *Dead and earthquake loads(Y direction)* | *Dead and earthquake loads(Z direction)* |
| Displacement(mm) | | 42.0 | 44.0 | 19.6 |
| Stresses (MPa) | Comp. *Outer* | 2.05 | 1.15 | 1.35 |
| | Comp. *Inner* | 1.85 | 1.10 | 1.25 |
| | Tension *Outer* | 1.05 | 0.95 | 0.85 |
| | Tension *Inner* | 1.00 | 0.75 | 0.80 |
| | Shear *Outer* | 0.65 | 0.60 | 0.50 |
| | Shear *Inner* | 0.60 | 0.55 | 0.45 |



**Table 4 Whole analyses results of the mosque with 8window openings case**

| Analyses Data | | Analyses | | |
|---|---|---|---|---|
| | | *Dead and earthquake loads(X direction)* | *Dead and earthquake loads(Y direction)* | *Dead and earthquake loads(Z direction)* |
| Displacement(mm) | | 36.0 | 39.0 | 9.80 |
| Stresses (MPa) | Comp. *Outer* | 1.74 | 1.00 | 1.15 |
| | Comp. *Inner* | 1.51 | 0.95 | 1.05 |
| | Tension *Outer* | 0.95 | 0.80 | 0.69 |
| | Tension *Inner* | 0.90 | 0.65 | 0.58 |
| | Shear *Outer* | 0.55 | 0.50 | 0.30 |
| | Shear *Inner* | 0.50 | 0.45 | 0.20 |

## 4 Conclusions

In this study restoration effects on the earthquake behaviour of masonry mosques is investigated with considering different opening ratios on dome. For this purpose Hüsrev Pasha Mosque was selected and finite element model of the mosque constituted with SAP2000 software. Two cases considered in analyses for to determine the restoration effects namely firstly the mosque constituted with 16 window openings then constituted with 8 window openings. In order to better understand the earthquake behaviour of the mosque, earthquake loads were affected from three directions, x, y, z. Mode Superposition Method was used in earthquake analyses. As a result of the study the following observations were made:

- To understand the structural behaviour of Hüsrev Pasha Mosque, firstly 16 then 8 window openings on dome taken into consideration in modelling and four different analyses were made for each case, those are modal analysis and earthquake analyses which were effected x, y and z directions.

- Modal analyses were made for to determine the dynamic characteristics of Hüsrev Pasha Mosque for each case and first 20 modes obtained. Damping ratio was taken as %5. Frequencies of the mosque vary from 3.74 Hz to 5.66 Hz for 16 window case and from 3.81 Hz to 5.78 Hz for 8 window case for first 4 modes. Mode shapes for each case are; translation x and y directions, squeeze and torsion, respectively. Analyses were made with using first 20 modes.

- When compared the results of modal analyses, reduction of window openings are caused an increase on frequencies. This situation shows that reduction of window openings affects the structural and stability of the mosque positively.

- Displacements are occurred at middle point of big dome as 42.0 mm and 44.0 mm for horizontal directions and 19.6 mm for vertical directions in the analyses with 16 window openings. Considering the height of the mosque, which is


15 m, maximum relative displacement at peak point of big dome is approximately 0.003. This value remains in acceptable limits.

- Displacements are occurred at middle point of big dome as 36.0 mm and 39.0 mm for horizontal directions and 9.8 mm for vertical directions in the analyses with 8 window openings. Considering the height of the mosque, which is 15 m, maximum relative displacement at peak point of big dome is approximately 0.0025. This value remains in acceptable limits. The displacement results show that, reduction of window openings from 16 to 8 reduces the displacements at a certain rate and this is good for the structural behaviour of the mosque.

- When the stress results are examined it is seen that compression and shear stresses values don't exceed the recommended compression and shear stresses values in Earthquake Resistant Design Rules for Masonry Structures in the Turkish Earthquake Code (2007).

- Tension stresses are occurred at near openings, bottom of walls and under the dome areas locally for each case. These values are in acceptable level for horizontal forces.

- Compression stresses, tension stresses and shear stresses collects some critic areas which are especially near openings and crossing points because of this, those areas must construct as monolithic in restoration process.

- When the displacement results are compared, %14, %11 and %50 decrease ratios are obtained for dead and earthquake analyses with x, y and z directions respectively.

- According to the comparison between the compression stresses results, decrease ratios are obtained as %15, %13 and %15 for dead and earthquake analyses with x, y and z directions respectively for outer side of the mosque and %18, %13 and %16 for dead and earthquake analyses with x, y and z directions respectively for inner side of the mosque.

- According to the comparison between the tensile stresses results, decrease ratios are obtained as %9, %16 and %19 for dead and earthquake analyses with x, y and z directions respectively for outer side of the mosque and %10, %13 and %27 for dead and earthquake analyses with x, y and z directions respectively for inner side of the mosque.

- When the shear stresses results are compared, decrease ratios are obtained as %15, %16 and %40 for dead and earthquake analyses with x, y and z directions respectively for outer side of the mosque and %16, %18 and %55 for dead and earthquake analyses with x, y and z directions respectively for inner side of the mosque.

- FRP strengthening can be used for to resist the tension stresses in restoration applications.

Consequently, the reduction of window openings ensures integrity on the walls, so this situation supports the structural performance of the mosque. It is seen from the study that restoration applications, especially reduction of the window openings on dome are improves the earthquake response of the mosque.





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
