# Peer review of "Earthquake Response of Heavily Damaged Historical Masonry Mosques after Restoration"

_Natural Hazards and Earth System Sciences, 2017_

## Referee Comment (RC1) · B. SEVÄřM (Referee) · 21 Apr 2017

Manuscript ID. : nhess-2017-141 (Natural Hazards and Earth System Sciences) Author (s) : Ahmet Can AltunÄśşÄśk and Ali Fuat Gençu Title : Earthquake Response of Heavily Damaged Historical Masonry Mosques after Restoration

GENERAL COMMNETS: This paper presents a detail investigation about the structural dynamic response of masonry mosques, which was nearly collapsed after earthquake, after restoration studies. This is very interesting, well written and organized paper. This paper will be very useful for academic researchers and project engineers related to this subject. The reviewer recommends the publication of the paper after minor revisions

given in below in Natural Hazards and Earth System Sciences.

SPECIFIC COMMENTS: • Fig. 1 should be removed. The earthquake region can be stated as one sentence in related places. • The section "Structural analyses of the mosque with 8 window openings" should be removed to avoid the repetition. It is shown that the results can be summarized in Tables 3 and 4 for better understanding. • The conclusion part of the paper should be shortened. Only main results should be given.

Best regards. . .

---

## Referee Comment (RC2) · Anonymous Referee #2 · 19 Jun 2017

This paper is suitable for publication in your journal.

---

## Author Comment (AC1) · 19 Jun 2017

Ms. Ref. No : nhess-2017-141-R1 (Natural Hazards and Earth System Sciences) Author(s) : Ahmet Can AltunÄśşÄśk and Ali Fuat Genç Title : Earthquake Response of Heavily Damaged Historical Masonry Mosques after Restoration

ANSWERS TO THE COMMENTS: First of all, we would like to thank the editor and reviewers for their helpful comments and contributions to our study. We have tried to answer the number of points that the editor and reviewers have outlined. These points are discussed below following the editor and reviewers' comments. We hope that the points amended in the paper comprehensively would be satisfactory enough

to correspond the comments of the editor and reviewers.

Comments: This paper presents a detail investigation about the structural dynamic response of masonry mosques, which was nearly collapsed after earthquake, after restoration studies. This is very interesting, well written and organized paper. This paper will be very useful for academic researchers and project engineers related to this subject. The reviewer recommends the publication of the paper after minor revisions given in below in Natural Hazards and Earth System Sciences.

Reviewer 1: Fig. 1 should be removed. The earthquake region can be stated as one sentence in related places.

Answer: According to the reviewer suggestions, Fig. 1 is deducted. This point is ordered in the revised paper.

Reviewer 1: The section "Structural analyses of the mosque with 8 window openings" should be removed to avoid the repetition. It is shown that the results can be summarized in Tables 3 and 4 for better understanding.

Answer: According to the reviewer suggestions, the section is deducted to better understanding and avoid the repetition. This point is ordered in the revised paper. Reviewer 1: The conclusion part of the paper should be shortened. Only main results should be given.

Answer: According to the reviewer suggestions, the conclusion part of the paper is revised as:

"In this study restoration effects on the earthquake behaviour of masonry mosques is investigated with considering different opening ratios on dome. As a result of the study the following observations were made:

• Reductions of window openings are caused an increase on frequencies. This situation shows that reduction of window openings affects the structural and stability of the mosque positively. • When the stress results are examined it is seen that compression and shear stresses values don't exceed the recommended compression and shear stresses values for masonry structures in the Turkish Earthquake Code (2007). • Tension stresses are occurred at near openings, bottom of walls and under the dome areas locally for each case. • Compression stresses, tension stresses and shear stresses collects some critic areas which are especially near openings and crossing points because of this, those areas must construct as monolithic in restoration process. FRP strengthening can be used for to resist the tension stresses in restoration applications. • According to the results, the displacements, compression, tension and shear stresses decrease when the openings reduced.

Consequently, the reduction of window openings ensures integrity on the walls, so this situation supports the structural performance of the mosque. It is seen from the study that restoration applications, especially reduction of the window openings on dome are improves the earthquake response of the mosque."

This point is ordered in the revised paper.

Please also note the supplement to this comment:
http://www.nat-hazards-earth-syst-sci-discuss.net/nhess-2017-141/nhess-2017-141-AC1-supplement.pdf

―――――――――――――

[Figure]

**Supplement:**

**Ms. Ref. No** : nhess-2017-141-R1 (*Natural Hazards and Earth System Sciences*)

**Author(s)** : Ahmet Can Altunışık and Ali Fuat Genç

**Title** : Earthquake Response of Heavily Damaged Historical Masonry Mosques after Restoration

**ANSWERS TO THE COMMENTS:**

First of all, we would like to thank the editor and reviewers for their helpful comments and contributions to our study. We have tried to answer the number of points that the editor and reviewers have outlined. These points are discussed below following the editor and reviewers' comments. We hope that the points amended in the paper comprehensively would be satisfactory enough to correspond the comments of the editor and reviewers.

**Comments:** This paper presents a detail investigation about the structural dynamic response of masonry mosques, which was nearly collapsed after earthquake, after restoration studies. This is very interesting, well written and organized paper. This paper will be very useful for academic researchers and project engineers related to this subject. The reviewer recommends the ***publication of the paper*** after minor revisions given in below in Natural Hazards and Earth System Sciences.

**Reviewer 1:** Fig. 1 should be removed. The earthquake region can be stated as one sentence in related places.

**Answer:** *According to the reviewer suggestions, Fig. 1 is deducted. This point is ordered in the revised paper.*

**Reviewer 1:** The section "Structural analyses of the mosque with 8 window openings" should be removed to avoid the repetition. It is shown that the results can be summarized in Tables 3 and 4 for better understanding.

**Answer:** *According to the reviewer suggestions, the section is deducted to better understanding and avoid the repetition. This point is ordered in the revised paper.*

**Reviewer 1:** The conclusion part of the paper should be shortened. Only main results should be given.

**Answer:** *According to the reviewer suggestions, the conclusion part of the paper is revised as:*

[revised manuscript text omitted]

---

## Author Comment (AC2) · 20 Jun 2017

Thanky you very much for your good comments.

---

## Author Response (AR2)

**Ms. Ref. No** : nhess-2017-141-Revision (*Natural Hazards and Earth System Sciences*)

**Author(s)** : Ahmet Can Altunışık and Ali Fuat Genç

**Title** : Earthquake Response of Heavily Damaged Historical Masonry Mosques after Restoration

**ANSWERS TO THE COMMENTS:**

First of all, we would like to thank the editor and reviewers for their helpful comments and contributions to our study. We have tried to answer the number of points that the editor and reviewers have outlined. These points are discussed below following the editor and reviewers' comments. We hope that the points amended in the paper comprehensively would be satisfactory enough to correspond the comments of the editor and reviewers.

**Comments:**

**Editor:** Dear authors, thanks for your revisions. Your manuscript is now fine in principle - BUT: You need to include another paragraph in which you acknowledge further international literature on that topic. This section is far too weak ... As soon as you have finalized that, I will perform a final check and if everything is fine, the paper will be published.

**Answer:** *According to the editor suggestions, a paragraph is added to the revised paper related to international literature on topic as:*

> *"There are many studies in the literature about historical masonry structures. Almost every aspect of the subject has been examined experimentally and numerically. Historical masonry arch bridges (Milani and Lourenço, 2012; Altunisik et al., 2015), towers (Peña et al., 2010), minarets and mosques (Seker et al., 2014), churches (Brandonisio et al., 2013), buildings and walls (Shariq et al., 2008; Lin et al., 2012; Parisi et al., 2013), chimneys (Minghini et al., 2014) etc. have all been investigated by different authors. Very rigid structures such as castles, fortresses and bastions have been investigated in terms of static and dynamic structural behavior (Betti et al., 2011; Tiberti et al., 2016)."*

References:
- Altunisik, A.C., Kanbur, B. and Genc, A.F.: The effect of arch geometry on the structural behavior of masonry bridges, Smart Struct Syst., 16, 1069–1089, 2015.
- Betti, M., Orlando, M. and Vignoli, A.: Static behaviour of an Italian Medieval Castle: Damage assessment by numerical modeling, Comput. Struct., 89, 2011.

- Brandonisio, G., Lucibello, G., Mele, E. and De Luca, A.: Damage and performance evaluation of masonry churches in the 2009 L'Aquila earthquake, Eng. Fail. Anal., 34, 693-714, 2013.
- Milani, G. and Lourenço, P.B.: 3D non-linear behavior of masonry arch bridges, Comput. Struct., 110, 133-150, 2012.
- Minghini, F., Milani, G. and Tralli, A.: Seismic risk assessment of a 50m high masonry chimney using advanced analysis techniques, Eng. Struct., 69, 255-270, 2014.
- Peña, F., Lourenço, P.B., Mendes, N. and Oliveira, D.V.: Numerical models for the seismic assessment of an old masonry tower, Eng. Struct., 32, 1466-1478, 2010.
- Seker, B.S., Cakir, F., Dogangun, A. and Uysal, H.: Investigation of the structural performance of a masonry domed mosque by experimental tests and numerical analysis, Earthquakes Struct., 6, 335-350, 2014.
- Tiberti, S., Acito, M. and Milani, G.: Comprehensive FE numerical insight into Finale Emilia Castle behavior under 2012 Emilia Romagna seismic sequence: damage causes and seismic vulnerability mitigation hypothesis, Eng. Struct., 117, 397-421, 2016.